# Comprehensive Molecular Analysis Identified an SRSF Family-Based Score for Prognosis and Therapy Efficiency Prediction in Hepatocellular Carcinoma

**DOI:** 10.3390/cancers14194727

**Published:** 2022-09-28

**Authors:** Jingsheng Yuan, Zijian Liu, Zhenru Wu, Jiayin Yang, Tao Lv

**Affiliations:** 1Department of Liver Surgery and Liver Transplantation Center, West China Hospital of Sichuan University, Chengdu 610041, China; 2Laboratory of Liver Transplantation, Frontiers Science Center for Disease-Related Molecular Network, West China Hospital of Sichuan University, Chengdu 610041, China; 3Department of Radiation Oncology, Cancer Center, West China Hospital of Sichuan University, Chengdu 610041, China; 4Laboratory of Pathology, Key Laboratory of Transplant Engineering and Immunology, NHC, West China Hospital of Sichuan University, Chengdu 610041, China

**Keywords:** hepatocellular carcinoma, alternative splicing, serine/arginine-rich splicing factors, prognostic model, therapeutic response

## Abstract

**Simple Summary:**

Hepatocellular carcinoma (HCC) remains one of the most common malignancies worldwide. Despite significant progress in the comprehensive management of HCC, continued efforts are still necessary to identify prognostic assessment approaches for patient-specific survival so that more appropriate treatment and management regimens can be proposed for different HCC subpopulations. Here, we comprehensively exploited the expression characteristics of 12 SRSF family members across multiple datasets. We further identified an SRSF score consisting of 18 SRSF-related genes which were associated with overall survival and drug sensitivity in HCC. Moreover, the predictive power of the SRSF score was validated in independent HCC cohorts and different HCC subgroups. We further explored the effect of SRSF11 expression on HCC cell proliferation and drug sensitivity. Overall, our study provides a novel predictive model to assess the prognosis and drug sensitivity of HCC.

**Abstract:**

The serine/arginine-rich splicing factors (SRSF)-mediated alternative splicing plays an essential role in the occurrence and progression of hepatocellular carcinoma (HCC). However, the SRSF-based signature that can predict the prognosis and therapy efficiency is yet to be investigated in HCC. Here, we comprehensively assessed the landscape and prognostic significance of the SRSF family genes in HCC. Then, we screened the SRSF family-related genes for signature construction and explored their biological characteristics. We further established an SRSF score consisting of 18 SRSF-associated genes and evaluated its correlation with prognosis and drug sensitivity in HCC. The predictive power of the SRSF score was validated in independent HCC cohorts and different HCC subgroups. Moreover, we further investigated that knockdown of SRSF11, a pivotal gene in the SRSF score, inhibited CDK1-dependent proliferation and enhanced the drug sensitivity of HCC cells. Overall, our study identified a novel SRSF family-based predictive model, and we demonstrated that SRSF11 is a promising therapeutic target for HCC, which enhances our understanding of the SRSF family genes and provides valuable insights into the clinical treatment and molecular mechanisms of HCC.

## 1. Introduction

Hepatocellular carcinoma (HCC) is the most common type of primary liver cancer (approximately 90%), and the fourth leading cause of cancer-related deaths worldwide [1,2]. Since 2015, major advances have been made in the comprehensive management of HCC, especially in the development of systemic therapies, which have dramatically improved the quality of life and overall survival rates of HCC patients [3]. Currently, despite the clinical improvements in the combination of therapeutic strategies based on histological subtypes and individualized therapy, the 5-year survival rate is still less than 20% due to high recurrence and metastasis rates [4,5]. Moreover, it is estimated that HCC will affect more than 1 million individuals annually by 2025 [6]. Therefore, continued efforts are still necessary to discover prognostic assessment approaches for patient-specific survival so that the most appropriate treatment and management regimens can be proposed for different HCC subpopulations.

Most therapeutic agents available for tumors target specific genes or genomes [7,8], but researchers realize that these specific genes or genomes contain only a limited number of targets. In recent years, with the advancement of multiomics technologies, many studies using diverse expression profiles and bioinformatics methods have provided additional prognostic assessment and drug–target predictions for HCC patients [9,10]. However, tumor genetic alterations are diverse and complex, and it is important to establish multigene classifiers to predict the treatment and prognosis of HCC. Therefore, it is necessary to identify novel biomarkers, or tumor-associated genomic profiles, to provide a multifaceted view of tumor characteristics for the exploration and development of new therapeutic strategies, potentially driving further identification and optimization of individualized treatment regimens for HCC patients.

Alternative splicing (AS) is an essential step in gene expression and regulation [11]. Moreover, AS leads to the generation of transcriptome and biological diversity in malignant tumor progression [12]. The serine/arginine-rich splicing factors (SRSF) belong to the serine-arginine-rich protein family, canonically consisting of 12 members (SRSF1–12) [13]. As key regulators of tumor biology, SRSF widely participate in regulation, ranging from RNA stability to RNA splicing and even RNA modification, export and translation [14]. Since most of the current studies have focused on a single SRSF protein, the understanding of the SRSF family is still limited [15]. For instance, do SRSF regulatory targets have something in common? Are the potential regulatory mechanisms of all these targets similar? Is it possible to target SRSF in tumorigenesis to achieve tumor therapy? Therefore, further identifying the functions and mechanisms of distinct SRSF in tumorigenesis will help to provide richer research insights and therapeutic strategies for malignant tumors.

Here, we exploited 12 SRSF family members and explored their expression characteristics across multiple datasets. Importantly, we identified an SRSF score composed of 18 SRSF-related genes, and this SRSF score was correlated with the overall survival and drug sensitivity of HCC. Moreover, the predictive power of the SRSF score was validated in independent HCC cohorts and different HCC subgroups. We further investigated the effect of SRSF11 expression on the proliferation and drug sensitivity of HCC cells. Overall, our study provides a novel predictive model to evaluate the prognosis and drug sensitivity of HCC.

## 2. Materials and Methods

### 2.1. Clinical Samples and Immunohistochemistry

Thirty fresh human HCC and paired noncancer liver tissues were collected from May 2014 to December 2020 at the West China Hospital of Sichuan University (Appendix A). All samples were confirmed to have a clinicopathological diagnosis of HCC through pathology reports. The study using clinical samples was approved by the Ethics Committee on Biomedical Research, West China Hospital of Sichuan University. Informed consent was obtained from all patients or their relatives. Additionally, immunohistochemistry (IHC) was performed as previously described [16]. The primary antibodies used in this study are listed in Appendix A.

### 2.2. Cell Culture and Reagents

Huh7, Hep3B, PLC5 and SNU387 cell lines were purchased from the National Collection of Authenticated Cell Cultures (Shanghai, China) and were cultured in complete medium containing Dulbecco’s modified Eagle’s medium (DMEM) (HyClone, UT, USA) supplemented with 10% fetal bovine serum (FBS) (Gibco, NY, USA), 1000 U/mL penicillin and 100  μg/mL streptomycin (HyClone, UT, USA), and were grown in a humidified air atmosphere containing 5% CO_2_ at 37 °C. Sorafenib-resistant HCC cell lines from Huh7 and HepG2 parental cells were cultured for 8 months, as previously described [16]. All cell lines were analyzed by STR profiling for cell line authentication, and routine mycoplasma detection was performed. Additionally, Sorafenib (S7397), roscovitine (S1153), doxorubicin (E2516), etoposide (S1225), gemcitabine (S1714), GDC-0941 (S1065) and SB590885 (S2220) were purchased from Selleckchem (Houston, TX, USA).

### 2.3. Transfection

Transfection was performed as previously described [16,17]. The short interfering RNAs (siRNAs) were constructed by Tsingke (Tianjin, China). Additional information about siRNAs is available in Appendix A.

### 2.4. Western Blot Analysis

Western blot analysis was performed as previously described [16,17]. The primary antibodies used in this study are listed in Appendix A.

### 2.5. Quantitative Real-Time Polymerase Chain Reaction

Quantitative real-time polymerase chain reaction (qRT–PCR) was performed as previously described [16,17]. The primers used in our study are listed in Appendix A.

### 2.6. Cell Counting kit-8 and EdU Assays

The cell counting kit-8 (CCK-8) was performed as previously described [18]. EdU assays were performed using a BeyoClick™ EdU Cell Proliferation Kit with Alexa Fluor 594 (Beyotime, Wuhan, China) according to the manufacturer’s instructions.

### 2.7. Data Sources and Preprocessing

The raw fragment per kilobase (FPKM) values and liver hepatocellular carcinoma (LIHC) clinical information in The Cancer Genome Atlas (TCGA) and International Cancer Genome Consortium (ICGC) datasets were downloaded from the UCSC XENA database. The raw data of FPKM values of noncancer liver tissues in Genotype-Tissue Expression (GTEx) datasets were also downloaded from the UCSC XENA database. Totally, 374 HCC and 50 noncancer liver samples were involved in the TCGA dataset, 243 HCC and 202 noncancer liver samples were contained in the ICGC dataset, while 110 noncancer liver samples were comprised in the GTEx dataset. After removing bias effects and unified disposal, we used “Combat” algorithm from R package “sva” to combine sequencing profiles from these three independent datasets [19]. Principal component analysis (PCA) was employed to evaluate the dimensions of the three datasets before and after data preprocessing. As a result, 617 HCC and 362 noncancer liver samples were involved in this study.

### 2.8. Functional and Pathway Enrichment Analysis

The Kyoto Encyclopedia of Genes and Genomes (KEGG) pathway and gene ontology (GO) analysis results were obtained from The Database for Annotation, Visualization and Integrated Discovery (DAVID) v6.8 [20] (https://david.ncifcrf.gov/tools.jsp accessed on 23 March 2022). The GO analysis contained three categories named biological process, molecular function and cellular component. To further estimate pathway and biological process activity variations in samples from expression datasets, we performed gene set variation analysis (GSVA) enrichment analysis using the “GSVA” R package, a nonparametric and unsupervised method [21].

### 2.9. Assessment of the Tumor Immune Microenvironment

As described above, we used the GSVA method to perform gene-set enrichment analysis to quantify the relative abundance of each infiltrating cell in a single sample. The immune cell markers used in this study were extracted from two previously published authoritative studies [22,23].

### 2.10. Establishment of the SRSF-Related Score

Correlation analyses were conducted among SRSF family members and protein coding RNAs in HCC. Then, univariate Cox regression analysis was used to observe the correlation between the expression level of these related genes and patient prognosis of HCC. Genes with *p* < 0.05 in Univariate Cox regression analysis were included in the construction of prognostic risk models. Then, a sliding windows sequential forward feature selection (SWSFS) algorithm was employed to further shrink the numbers of SRSF-related genes with prognostic value [24]. Next, the iteration least absolute shrinkage and selection operator (LASSO) Cox regression model was used to screen for the best gene signature in HCC [25]. Finally, the SRSF-related score was constructed with coefficients of members involved in the signature, and the median of the SRSF-related score was used as a cutoff to stratify patients into high- and low-SRSF score groups. The SRSF-related score could be calculated using the following formula: SRSF score = Σ (Coef i × Exp i), where i is the members involved in the gene signature. Differences in survival between the high- and low-SRSF score groups were further compared using Kaplan–Meier analysis. Stratification analysis was performed to test whether the SRSF-related score was an independent prognostic factor in HCC. Using the “timeROC” package in R, the area under the curve (AUC) was used to test the performance of the classifier. We also assessed the prognostic value of these genes in the model.

### 2.11. Development and Validation of the Prognostic Nomogram

Based on the risk score of clinical risk factors and multivariate Cox regression coefficients, a prognostic nomogram was built using the “rms” R package, and the predictive accuracy of this nomogram was assessed using the calibration curve and the concordance index.

### 2.12. mRNA-Based Stemness Index (mRNAsi) and Therapeutic Response Prediction

To assess the stemness of cancer cells, a one-class logistic regression algorithm, mRNAsi, was used to calculate the stemness index for each HCC sample using the workflow available on a previously established database [26]. The therapeutic response for each sample was predicted according to the largest public pharmacogenomics database, the Genomics of Drug Sensitivity in Cancer (GDSC) (https://www.cancerrxgene.org/ accessed on 23 March 2022). We used the R package “pRRophetic” to predict the samples’ half-maximal inhibitory concentration (IC50), following the instructions described previously [27]. The prediction process was implemented based on ridge regression and the prediction accuracy was evaluated by a 10-fold cross-validation based on the GDSC training set.

### 2.13. Statistical Analysis

All statistical calculations were performed using R software (version 3.6.1, R Foundation for Statistical Computing, Vienna, Austria). To calculate the tumor mutational burden (TMB) per megabase, the total number of mutations counted was divided by the size of the coding region of the targeted territory in the TCGA-LIHC cohort [28]. The comparison of normally distributed variables between the two groups was performed using an unpaired *t*-test, and the statistical significance of the nonnormally distributed variables was estimated using the Mann–Whitney U test (Wilcoxon rank-sum test). Spearman’s correlation analysis was performed to calculate the correlation coefficient between the two factors. Based on the correlation between gene expression and patient survival, the optimal cutoff point for each dataset was determined using the “survminer” R package, and the “surv-cutpoint” function was used to repeat all potential cutoff points to obtain the maximum rank statistic, divided into two groups: high and low. Survival curves for prognostic analysis were generated using the Kaplan–Meier method, and significant differences were determined using the log-rank test. The false discovery rate (FDR) method was used to adjust the P value for multiple comparisons, and statistical significance was set at *p* < 0.05; that is, the FDR was less than 0.05. The asterisks represent the statistical *p*-value (* *p* < 0.05; ** *p* < 0.01; *** *p* < 0.001; ns, no significant).

## 3. Results

### 3.1. Landscape and Prognostic Significance of the SRSF Family Genes in HCC

We enrolled 12 well-defined SRSF family genes in this study. Raw data of HCC and normal liver tissue from the GTEx, TCGA and ICGC databases were downloaded and used for subsequent analysis. To increase the credibility of the analysis results, we first expanded the sample size from various public datasets. Notably, we found that the dimensions of the three datasets were not well unified before data preprocessing (Figure 1A). Thus, we used the “Combat” algorithm from the R package “sva” to eliminate the batch effects from nonbiological technical biases among different testing cohorts [19]. The PCA results further revealed that the combined expression profiles after normalization possessed stronger comparability than the raw data (Figure 1B). For simplicity and the credibility of subsequent analysis, we mainly used these combined gene expression profiles, named the aggregated HCC cohort, for further analysis.

To reveal the landscape of SRSF family genes between HCC and normal liver tissues, we first investigated the differential distribution and copy number variation of SRSF family members by integrating bioinformatics analysis. Abundance of the SRSF family genes is displayed in the aggregated HCC cohort, indicating that most SRSF family members were highly expressed in HCC compared to normal liver tissues (Figure 1C). To further validate the expression differences of SRSF family genes, we further performed pairwise analysis on 50 pairs of samples from the TCGA dataset and 199 pairs of samples from the ICGC dataset, and the results indicated that eight genes (SRSF1, SRSF2, SRSF3, SRSF6, SRSF7, SRSF9, SRSF11, SRSF12) were markedly more abundant in HCC than in normal liver tissue (Figure 1D,E). By analyzing the somatic copy number landscape of the SRSF family members in HCC, we spotted copy number variation (CNV) in the SRSF genes, but only in a small proportion of HCC samples (Figure 1F). A comparative analysis further suggested that the gene expression of SRSF2, SRSF6, SRSF9, SRSF11 and SRSF12 with CNV gain in tumors was significantly higher than that in the HCC without CNV gain samples (Figure 1G), which partially explained the abnormally elevated expression of these genes in HCC. 

We further evaluated the relationship between the expression of SRSF family genes and the OS and progression-free survival (PFS) of HCC patients utilizing univariate Cox proportional hazards regression analysis. Here, we found that 10 genes in the SRSF family were strongly associated with the OS of HCC patients. Moreover, all 10 genes were identified as “high-risk” factors of the OS, with a hazard ratio greater than 1 (Figure 2A). Likewise, we further confirmed that these 10 SRSF family genes were predictors of PFS in HCC (Appendix A). We further analyzed the gene expression landscape of SRSF family genes according to patient survival outcome and tumor stage. The results indicated that 7 genes in the SRSF family were positively correlated with the survival outcome of HCC patients (Figure 2B), and 8 genes in SRSF family were positively associated with tumor stage in HCC (Figure 2C). The expression correlation between the SRSF family members is illustrated in Figure 2D. Most of the genes exhibited a strong positive correlation with each other. The strong stratification power of SRSF family genes in predicting the OS of patients with HCC prompted us to further investigate SRSF family gene-related signaling pathways. Interestingly, GSVA revealed that these genes were tightly involved in proliferation-related biological processes, including DNA replication, E2F targets, the G2/M checkpoint, the mitotic spindle, the PI3K/AKT signaling pathway and the TGF-β signaling pathway. In contrast, SRSF family members were significantly negatively correlated with metabolism-related biological processes, including adipogenesis, bile acid metabolism and fatty acid metabolism (Figure 2E). The above results unequivocally indicated the crucial roles of SRSF family members in the occurrence and progression of HCC.

### 3.2. Identification of SRSF Family-Related Genes and Prediction of Their Functional Annotations in HCC

To construct a signature based on SRSF family-related genes, we performed a correlation analysis between SRSF family members and the gene expression profiles from the aggregated HCC cohort. The results suggested that, based on the criteria of a correlation coefficient greater than 0.6, the number of SRSF1-related genes was the highest and more than 400, while the number of SRSF12-related genes was the lowest, containing only 1 (Figure 3A and Appendix A). We named this group of genes (874 genes) as the SRSF family-related signature. Moreover, we presented the specific intergenic correlation networks between SRSF and the strong related genes with a correlation >0.7 in the upper right of Figure 3A, where we found most of the highly correlated genes linked to SRSF11 and SRSF1. Then, we analyzed the SRSF family-related signature using GO and KEGG analyses. The results of GO analysis showed that the SRSF family-related signature was involved in several cell division-related biological processes and molecular functions (Figure 3B,C). Moreover, the SRSF family-related signature mainly functions in the nucleus and chromatin region (Figure 3D). Likewise, the KEGG analysis showed that the SRSF family-related signature was primarily involved in the signaling pathways associated with the spliceosome, cell cycle and nucleocytoplasmic transport (Figure 3E), which was consistent with the GO findings. These results demonstrated that the SRSF family-related signature was critical for the regulation of the cell cycle and proliferation in HCC.

### 3.3. Construction of the SRSF score and Its Predictive Effect on the Prognosis of HCC Patients

To further endow the SRSF family-related signature with clinical significance and practicality, we attempted to construct a SRSF score. We first performed univariate Cox regression analysis for each gene in the SRSF family-related signature to screen and obtain 606 genes with prognostic implications in HCC (Appendix A). Then, as shown in Figure 4A, we employed Ranger, a weighted version of random forest based on the SWSFS algorithm, to evaluate the importance of each gene expression for prognostic. Ranger could provide a variable importance score for each SRSF family-related gene with an “out of bag (OOB)” error rate, when candidate genes were included one by one based on their variable importance scores ranks in the top of the training cohort [24]. After processing with the SWSFS algorithm, the results showed that when the number of genes was 34, the OBB error rate was the lowest, indicating that its predictive ability was the strongest. Thus, the top 34 SRSF-related genes were identified in the aggregated HCC cohort. We further presented the importance of these 34 genes in the upper right of Figure 4A and Appendix A. Next, we applied iterative LASSO Cox regression analysis to obtain 18 genes with independent prognostic significance in patients with HCC. Moreover, using the TCGA training cohort, we confirmed that these 18 genes could well predict the prognosis of HCC patients with an AUC of 0.761 and a hazard ratio of 3.78 (Figure 4B). Then, the SRSF score was calculated based on the regression coefficients and expression values of these 18 genes (Figure 4C and Appendix A). The SRSF score allowed patients to be divided into high-SRSF (*n* = 184, score value >0.955) and low-SRSF (*n* = 184, score value <0.955) score groups based on the median value. We found that the number of deaths of the HCC patients increased with the elevated SRSF score, reflecting a markedly higher mortality and shorter survival time in the high SRSF score group than in the low SRSF score group (Figure 4D). Moreover, the time-dependent AUC values of the SRSF score for the prediction of 1- to 8-year survival rates were all more than 0.7 (Figure 4E), indicating that the prediction accuracy is extremely high. These findings indicated that our SRSF score is an unbiased prognostic model that can be used as a reference tool for predicting the OS probability of HCC.

### 3.4. Validation of the Prognostic Predictive Capability of the SRSF Score in Different Clinical Subgroups

To further facilitate the clinical application of the SRSF score, a nomogram capable of predicting the 3- or 5-year survival probability of HCC patients was finally constructed (Figure 5A). The calibration curves at 3 and 5 years indicated good consistency between the prediction by the nomogram and actual overall survival outcomes (Figure 5B). To further validate the stability of the SRSF score in different clinical subgroups, we evaluated the predictive ability of the SRSF score in HCC patients with different ages. All patients were grouped by age and then ranked by the SRSF score into high- and low-SRSF score subgroups. Kaplan–Meier survival analyses were then used to estimate the difference in the OS of HCC patients between high- and low-score subgroups. Our results indicated that the OS in the high-score subgroup was markedly shorter than that in the low-score subgroup (Figure 5C). Since patients with early- (clinical stage I/II) or advanced- (clinical stage III/IV) stage HCC require different therapy strategies and exhibit different prognoses [6,7], we further applied the SRSF score to the patients with a clinical stage in the aggregated HCC cohort. Similarly, a high SRSF score was correlated with dramatically worse OS, regardless of whether the patient exhibited early- or advanced-stage HCC (Figure 5D). Likewise, we also found that regardless of HCC belonging to P53 wild-type (WT), P53 mutation (MUT), CTNNB1 WT or CTNNB1 MUT, the SRSF score provided statistically significant OS stratification (Figure 5E).

Furthermore, we also validated the prognostic predictive power of the SRSF score using the ICGC dataset. Kaplan–Meier survival curves of the ICGC dataset also indicated that HCC patients with high SRSF scores had a worse OS than those with low SRSF scores (Figure 5F), with a time-dependent AUC value of approximately 0.65 at 1 to 4 years (Figure 5G). A consistent Kaplan–Meier survival analysis result was obtained from the aggregated HCC data (Figure 5H). Moreover, the time-dependent AUC values of the SRSF score for the prediction of 3-, 5- and 8-year survival rates in the aggregated HCC data all exceeded 0.69 (Figure 5I). These data demonstrate that the SRSF score is stable and reliable in predicting the prognosis of HCC.

### 3.5. Identification of SRSF Score-Related Biological Characteristics

The strong stratification power of the SRSF score in predicting the OS probabilities of patients with HCC motivated us to further investigate the SRSF score-related biological characteristics. We first performed a GSVA, which consistently revealed that cell division-related pathways, PI3K/AKT/mTOR signaling pathways and P53 signaling pathways were highly activated in HCC patients with high SRSF scores, while metabolism and immune-related pathways were highly activated in HCC patients with low SRSF scores (Figure 6A). We then detected a positive correlation between the SRSF score and the previously reported mRNAsi index in the aggregated HCC data (Figure 6B), implying that the SRSF score was markedly associated with tumor stemness in HCC. Moreover, we also further confirmed the relationship between SRSF score and TMB, and the results indicated that HCC patients with high SRSF scores had a remarkably elevated TMB (Figure 6C). Given the great clinical promise of immunotherapy in the treatment of HCC, we further investigated the relationship between the SRSF score and immune cell infiltration in HCC patients. Notably, two previously published authoritative studies [22,23], which could distinguish 23 immune cell subtypes, were used to assess immune cell infiltration. The distribution of different immune cell subtypes between the high- and low-SRSF score groups are depicted in Figure 6D,E. Specifically, the low-SRSF score group had a higher population of B cells, NK cells, neutrophils, dendritic cells (DCs) and cytotoxic cells. Interestingly, HCC with a low-SRSF score had a higher proportion of activated functional cells, such as activated DCs and CD56-positive NK cells. We further demonstrated the relevance of SRSF family members to the abovementioned immune cells (Figure 6F).

### 3.6. The SRSF Score Predicts the Drug Sensitivity of HCC

Since previous data suggested a strong association between the SRSF score and cell proliferation-related signaling pathways, we intended to utilize the GDSC dataset to further investigate the relationship between the SRSF score and drug sensitivity. We first tested whether the SRSF score could predict the sensitivity of current first-line targeted drugs to HCC. Interestingly, the predictive value of sorafenib IC50 in HCC patients with low-SRSF scores was relatively weaker than that in HCC patients with high-SRSF scores (Figure 7A). We further analyzed the correlation between SRSF family members and sorafenib sensitivity utilizing the GDSC database, and the results suggested that SRSF11 had the strongest positive correlation with sorafenib IC50 (Figure 7B). We next screened inhibitors targeting SRSF score-related signaling pathways from the GDSC database, including the inhibitors of the cell cycle, PI3K/AKT/mTOR signaling pathway, P53 signaling pathway, and MAPK signaling pathway. We then evaluated the effects of these inhibitors in HCC cohorts in the GDSC database, and further estimated the IC50 values to explore the relationship between the SRSF score and inhibitor sensitivity of these targets. Unexpectedly, the analysis results indicated that 26 inhibitors were sensitive to HCC with low-SRSF scores, and 10 inhibitors were sensitive to HCC with high-SRSF scores (Figure 7C). The Sankey diagram in Figure 7D shows the flow/changes in the SRSF score with response to treatment.

### 3.7. SRSF11 Knockdown Inhibited the CDK1-Dependent Proliferation of HCC Cells

Using the Human Protein Atlas, we confirmed that eight previously identified differentially expressed genes had higher protein levels in HCC tissues than in noncancer liver tissues (Figure 7E and Appendix A). Notably, three SRSF family members were included in the SRSF score (Figure 4C). Since SRSF1 and SRSF2 have been previously reported to be involved in promoting the progression of HCC [29,30], our study implied the strongest positive correlation of SRSF11 with sorafenib IC50 (Figure 7B). Thus, we further validated the potential role of SRSF11 in HCC. Interestingly, our clinical samples highlighted a higher level of SRSF11 protein expression in HCC tissues than in noncancer liver tissues (Figure 8A). Moreover, SRSF11 expression was correlated with tumor diameter and tumor number (Appendix A). The correlation analysis of the TCGA-LIHC cohort also suggested that SRSF11 expression was associated with gender, lymph node metastasis, and tumor differentiation (Appendix A). We next transiently transfected HCC cells with SRSF11 siRNAs. The efficiency of transfection was demonstrated by qRT–PCR and western blotting results (Figure 8B–D). The original uncropped western blot images are shown in Appendix A. As expected, the results of CCK-8 and EdU assays suggested that knockdown of SRSF11 markedly suppressed the proliferation of HCC cells (Figure 8E,F). CDK1 is pivotal in regulating the G2-phase transition in the cell cycle, while CDK4 governs the commitment of eukaryotic cells to transition through the G1 phase to enter the DNA synthesis S phase [31]. Interestingly, our western blotting results indicated that SRSF11 knockdown remarkably inhibited CDK1 protein expression but not CDK4 (Figure 8C,D). These data imply that SRSF11 knockdown inhibited the CDK1-dependent proliferation of HCC cells.

### 3.8. SRSF11 Knockdown Enhanced the Drug Sensitivity of HCC Cells

We next investigated whether the expression level of SRSF11 affects the previously predicted inhibitor sensitivity. We first confirmed a positive correlation between the expression of SRSF11 and the predicted IC50 of sorafenib (Figure 9A). More importantly, our qRT-PCR results suggested that SRSF11 expression was significantly higher in previously cultured sorafenib-resistant HCC cells than in parental HCC cells (Figure 9B). We then further analyzed the correlation of SRSF11 with the sensitivity of the aforementioned inhibitors using the GDSC database. Interestingly, this result was generally consistent with the correlation between the SRSF score and the sensitivity of these inhibitors (Figure 9C). We further screened several inhibitors for critical targets and exhibited the potential IC50 of these inhibitors at various SRSF11 expression levels (Figure 9D). We next measured the IC50 of these inhibitors in HCC cells with various expression levels of SRSF11 utilizing CCK8 assays. Roscovitine is a selective CDK inhibitor targeting CDK1, but has little effect on CDK4/6 [32]. Our results suggested that the IC50 of roscovitine was markedly lower in SRSF11-suppressed HCC cells than in control HCC cells (Figure 9E). Consistent results were also obtained from the IC50 differences of the two inhibitors targeting DNA replication (doxorubicin and etoposide) and an inhibitor targeting nucleic acid synthesis (gemcitabine) in different SRSF11-expressing HCC cells (Figure 9F–H). GDC-0941 is a potent PI3Kα/δ inhibitor, while SB590885 is a potent B-Raf inhibitor [33,34]. Our CCK-8 results also verified that SRSF11 knockdown could remarkably enhance the sensitivity of HCC cells to these two pathway inhibitors (Figure 9I,J). These data suggest that SRSF11 knockdown contributes to the enhancement of systemic treatment in HCC.

## 4. Discussion

With the advancement of omics technologies over the past decade, an increasing number of tumor therapeutic targets and prognostic markers have been continuously defined [34,35]. Common genetic mutations, single nucleotide polymorphisms, genetic modifications, and posttranscriptional regulation have all been extensively studied, and this has indeed increased our understanding of cancer [36]. However, practical and reliable clinical applications are still very rare, suggesting that efforts are still needed to continue to explore tumor genetics and heterogeneity.

In recent years, the pivotal mechanisms of SRSF family genes involved in tumor proliferation, metastasis and drug resistance have been gradually discovered [29,31,37]. Therefore, investigating SRSF family-related genes may provide a new perspective for tumor therapy; however, the genetic landscape of the SRSF family in HCC has not been widely discussed. In this work, we comprehensively identified the expression landscape and prognostic features of 12 SRSF family members in HCC. Additionally, we constructed a SRSF score containing 18 SRSF family-associated genes to predict clinical prognosis and drug sensitivity in patients with HCC. More importantly, the validated reliability and the lesser gene composition of the SRSF score implies a promising clinical application in the future.

AS is a ubiquitous mechanism of gene expression regulation that permits the production of multiple distinct mRNA species from a single gene [38]. Some spliced mRNA isoforms may alter the reading frame, resulting in diverse protein isoforms with different functions or localizations [39]. However, not all detected AS events necessarily lead to the production of functional proteins, such as transcripts that are noncoding and not translated into proteins [40]. This gene-determining role is why we focused on the role of SRSF family genes in the progression of HCC. Among the approximately 200 proteins, there are two well-studied families of RNA splicing factors: heterogeneous nuclear ribonucleoprotein (hnRNP) proteins and SRSF family proteins [13]. Interestingly, our study found that SRSF family members are mainly involved in cell division-related tumor progression in HCC. However, no obvious mutation of SRSF family genes was found. Moreover, SRSF family members are only marginally associated with tumor metastasis and immunotherapy, so our study did not include this part of the mined data in the main text or validate it in the cytology experimental part. Notably, these data also imply that the AS-relevant regulatory mechanisms involved in HCC metastasis or the immune microenvironment may be correlated with other AS families or factors.

Among the SRSF scores, the pro-HCC effects of SRSF1 and SRSF2 have been previously reported [29,30]. Moreover, Lei et al. reported that SRSF1 promotes the inclusion of exon 3 of SRA1 and the invasion of hepatocellular carcinoma cells by interacting with exon 3 of SRA1pre-mRNA [41]. Luo et al. found that SRSF2 expression promoted cell proliferation and tumorigenic potential by controlling the expression of multiple gene variants [42]. SRSF11 has also been previously identified to be involved in cell cycle-specific recruitment of telomerase to telomeres at nuclear speckles, thereby promoting breast cancer progression [43]. However, the potential role of SRSF11 in HCC has not been fully elucidated. Therefore, our study further validated the function of SRSF11 in HCC cells. As expected, a high expression of SRAF11 markedly promoted DNA replication and cell proliferation in HCC cells, which in turn suggested that our SRSF11 score indeed has predictive power.

Previous studies have implied the possibility of SRSF family genes as potential therapeutic targets. Tam et al. reported that the CLK inhibitor SM08502 induces anti-tumor activity and reduces Wnt pathway gene expression by inhibiting SRSF phosphorylation and disrupted spliceosome activity in gastrointestinal cancer [44]. Lv et al. found that the reduction of SRSF1 promotes autophagy, which is critical to inhibiting Gefitinib-resistant lung cancer cell progression [45]. Because of the tight correlation between SRSF score and cell division-related signaling pathways, we further analyzed the association of the SRSF score with drug sensitivity using the GDSC database. Interestingly, our results indicated that SRSF scores can predict HCC sensitivity to multiple DNA replication inhibitors, CDK inhibitors and nucleic acid inhibitors. In addition to verifying the sensitivity of SRSF11 to some of the drugs mentioned above, we also validated the sensitivity of inhibitors of the PI3K/AKT signaling pathway and BRAF targets in the cytology assays, and the data were all consistent with the predicted results of the SRSF score. Finally, we also confirmed the relationship between SRSF11 expression and sorafenib sensitivity using clinical samples and cellular CCK-8 assays, and all these data strongly supported the potential value of SRSF score for predicting drug sensitivity in HCC.

The SRSF score is also somewhat restrictive. Our study found that the SRSF score had a strong positive correlation with the infiltration of T cells, NK cells and macrophages, which initially elaborated the relationship between the SRSF score and the immune microenvironment. Additionally, TMB has been reported to be associated with tumor immunotherapy efficacy. Our study also exhibited a positive correlation between the SRSF score and TMB in HCC. However, as mentioned above, no obvious association of the SRSF score with PD1 or PD-L1 was found in our analysis. Meanwhile, we failed to find a statistically significant relationship between the SRSF score and immunotherapy effects using the previously published TIDE algorithm. Thus, we next need to further expand the number of AS molecules or investigate the relationship between other AS family genes and immunotherapy.

## 5. Conclusions

In conclusion, this is the first and most comprehensive study of the expression profile landscape and clinical significance of SRSF family genes in patients with HCC. We also constructed a novel SRSF family gene-based signature, termed the SRSF score, to predict prognosis and drug sensitivity in HCC. These findings proposed a promising prognostic and therapeutic prediction model for HCC patients, providing a clinically useful tool for better prognostic management and optimization of associated treatments for HCC patients.

## Figures and Tables

**Figure 1 cancers-14-04727-f001:**
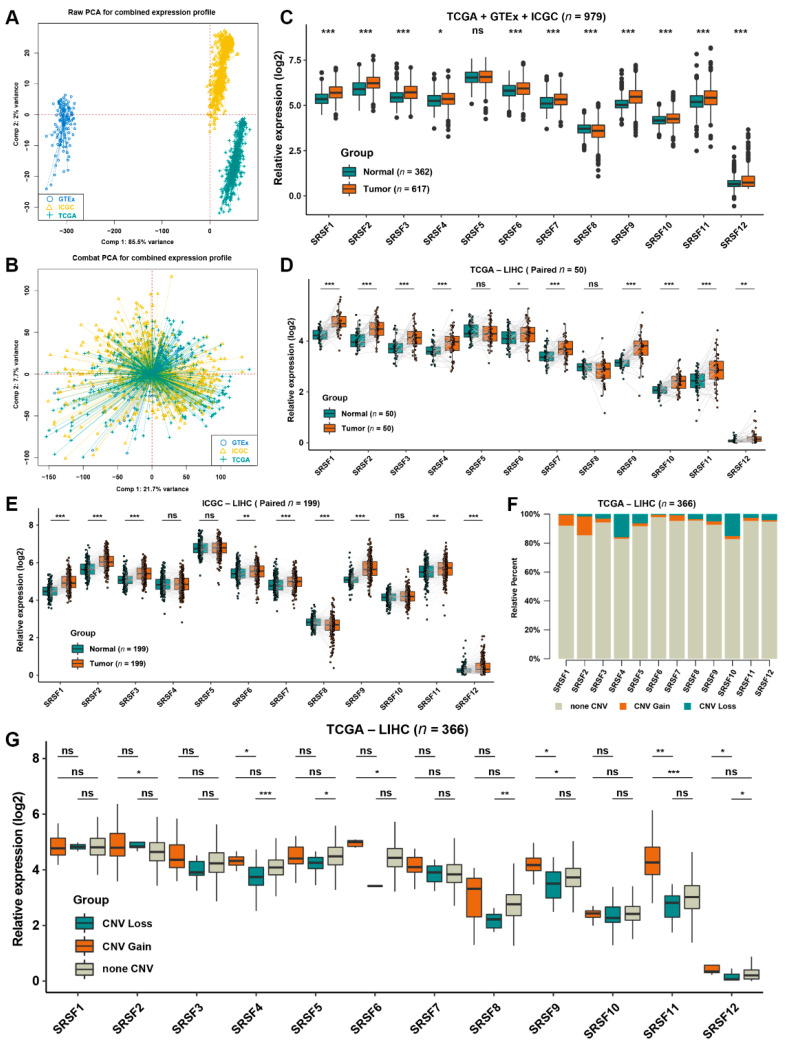
The expression pattern of SRSF family genes in HCC. (**A**) Principal component analysis (PCA) of the transcriptome profiles in distinct LIHC cohorts from the TCGA, ICGC and GTEx databases. (**B**) PCA of the transcriptome profiles of the aggregated HCC cohort. (**C**) The relative expression of 12 SRSF family genes in the aggregated HCC cohort. (**D**,**E**) The relative expression of 12 SRSF family genes in paired HCC and nontumor liver tissues from the (**D**) TCGA-LIHC and (**E**) ICGC-LIHC cohorts. (**F**) CNV of 12 SRSF family genes in HCC from the TCGA-LIHC cohort. (**G**) The relative expression of 12 SRSF family genes with different CNV states in the TCGA-LIHC cohort. The asterisks represent the statistical *p*-value (* *p* < 0.05, ** *p* < 0.01 and *** *p* < 0.001. ns, no significance).

**Figure 2 cancers-14-04727-f002:**
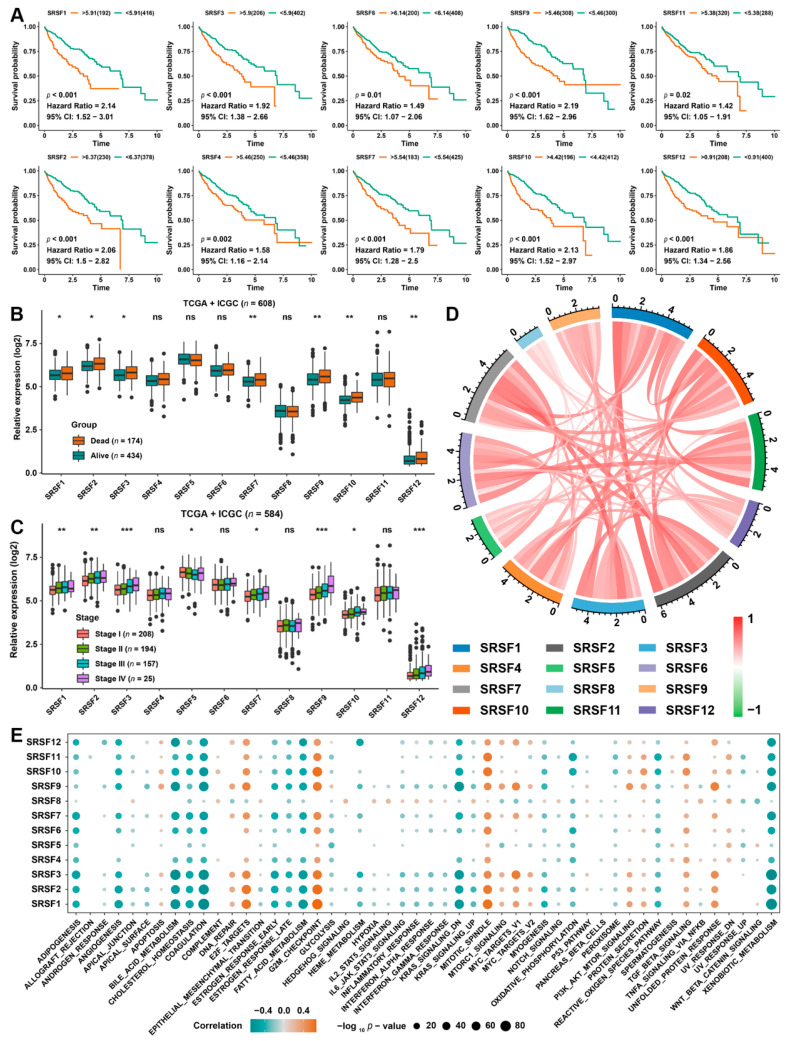
The prognostic analysis of SRSF family genes in HCC. (**A**) Kaplan–Meier curves of OS in the aggregated HCC cohort are stratified by the optimal cut-off value of the indicated SRSF genes. (**B**) The relative expression of 12 SRSF family genes based on the patient outcome in the aggregated HCC cohort. (**C**) The relative expression of 12 SRSF family genes based on tumor stage in the aggregated HCC cohort. (**D**) The interaction among SRSF family genes in HCC. The lines linking genes showed their interactions, and the thickness of lines showed the correlation strength. Positive correlations are marked with red lines, and negative correlations are marked with green lines. (**E**) KEGG signaling pathways associated with each SRSF family gene. The size of the circle represents the P value, with red representing a positive correlation and green representing a negative correlation. The asterisks represent the statistical *p*-value (* *p* < 0.05, ** *p* < 0.01 and *** *p* < 0.001. ns, no significance).

**Figure 3 cancers-14-04727-f003:**
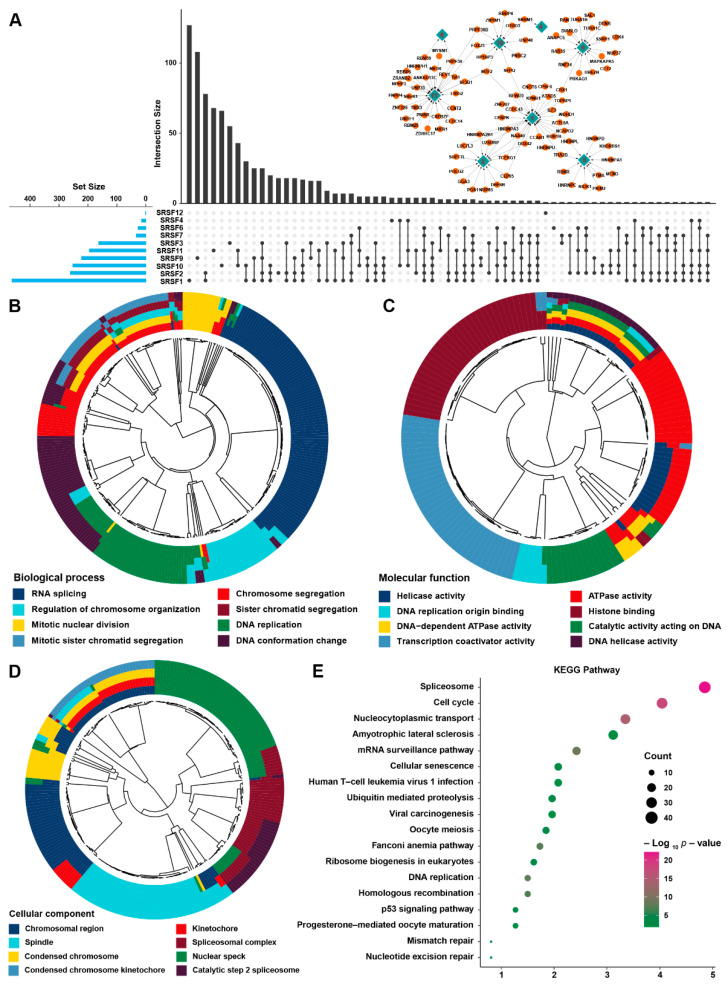
Construction of the SRSF family-based signature and its biological function analysis. (**A**) Venn diagram showing the specific or intersecting genes correlated with SRSF family members. The column on the left shows the sum of genes correlated with one of the SRSF family members, while the upper column shows intersection genes correlated with the SRSF family members. The specific correlation network containing genes with coefficients greater than 0.7 is shown in the upper right. (**B**–**D**) The enriched (**B**) biological processes, (**C**) molecular functions and (**D**) cellular components from GO analysis of the SRSF family-based signature in the aggregated HCC cohort. (**E**) KEGG pathway enrichment analysis of the SRSF family-based signature in the aggregated HCC cohort.

**Figure 4 cancers-14-04727-f004:**
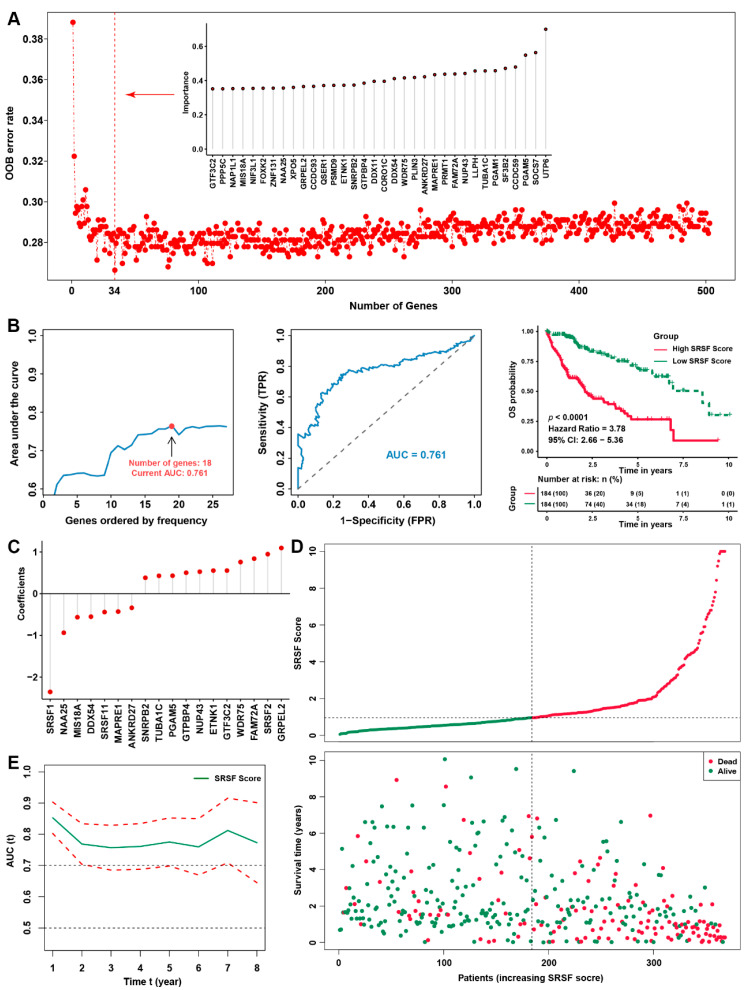
Construction of the SRSF score and the validation of its prognostic predictive value. (**A**) The relative importance of each gene of the SRSF family-based signature was calculated by random survival forest analysis, and the 34 most critical genes are shown. (**B**) Iteration LASSO analysis constructed an SRSF score with 18 genes. The AUC value was 0.761, and the survival curve for the SRSF score is shown in the right panel. (**C**) Coefficients of 18 genes involved in the SRSF score. (**D**) The tendency for the survival state of each HCC patient to change with the SRSF score. (**E**) The time-dependent AUC values of the SRSF score for the prediction of 1- to 8-year survival rates in the TCGA-LIHC dataset.

**Figure 5 cancers-14-04727-f005:**
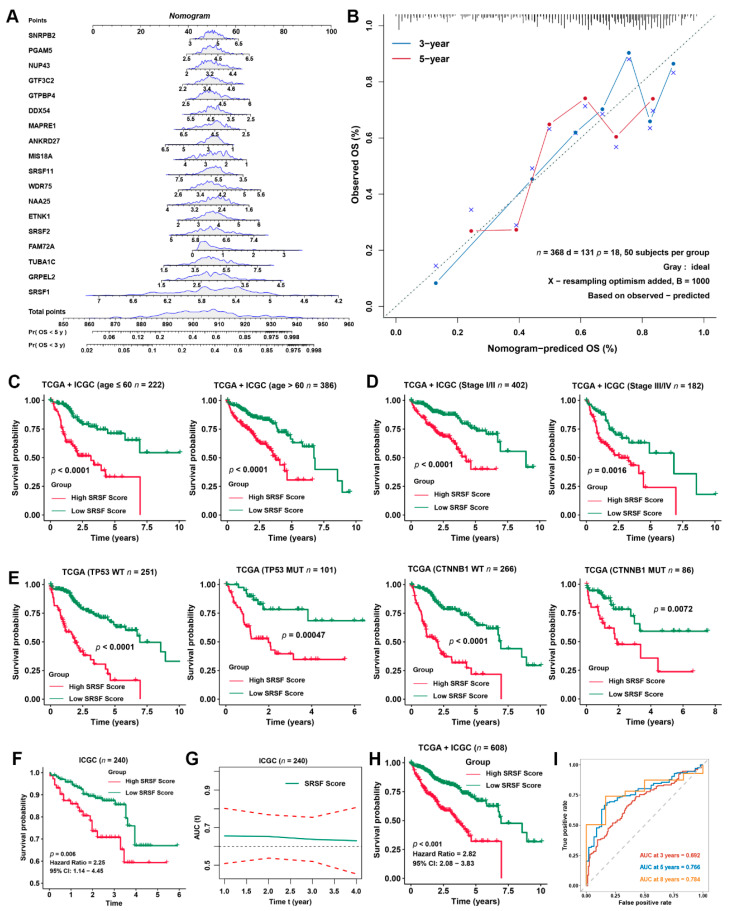
Validation of the prognostic predictive power of the SRSF score for HCC patients from different subgroups. (**A**) Nomogram to predict the 3- and 5-year overall survival of HCC patients. (**B**) Calibration curve for the comprehensive survival nomogram model in the TCGA-LIHC cohort. The dashed diagonal line represents the ideal situation, and the blue and red lines represent the 3- and 5-year observed nomograms, respectively. (**C**) Kaplan–Meier curves of the OS in HCC patients with different ages stratified by the SRSF score. (**D**) Kaplan–Meier curves of the OS in HCC patients with different tumor stages stratified by the SRSF score. (**E**) Kaplan–Meier curves of OS in HCC patients with different tumor mutation statuses stratified by the SRSF score. (**F**) Kaplan–Meier curves of OS in the ICGC-LIHC cohort stratified by the SRSF score. (**G**) The time-dependent AUC values of the SRSF score for the prediction of 1- to 8-year survival rates in the ICGC-LIHC cohort. (**H**) Kaplan–Meier curves of OS in the aggregated HCC cohort stratified by the SRSF score. (**I**) The time-dependent AUC values of the SRSF score for the prediction of 3-, 5- and 8-year survival rates in the aggregated HCC cohort.

**Figure 6 cancers-14-04727-f006:**
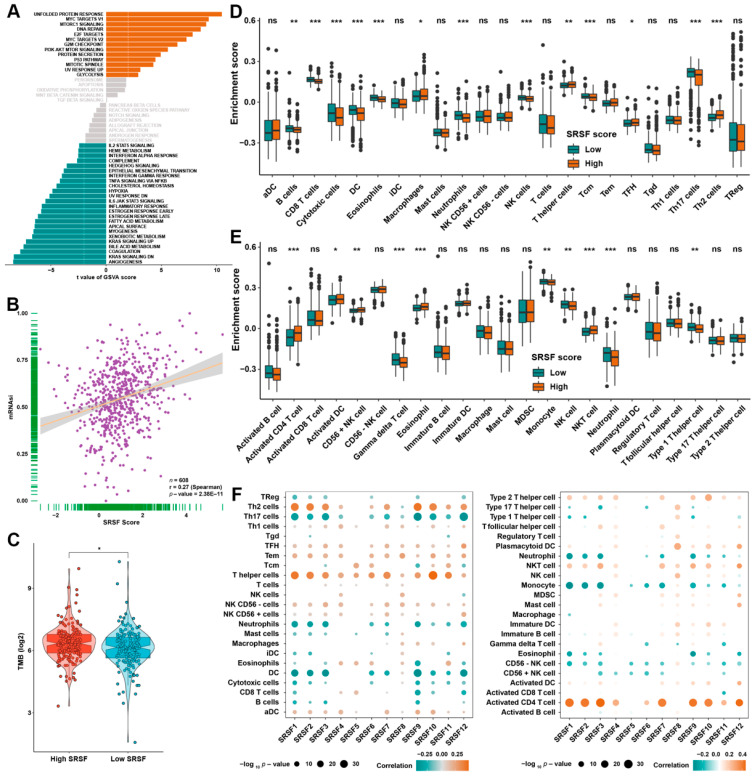
The biological features of HCC patients with various SRSF scores in the TCGA cohort. (**A**) GSVA showed the enriched signaling pathways in patients with different SRSF scores in the aggregated HCC cohort. (**B**) Correlation analysis between the SRSF score and the mRNAsi index. (**C**) TMB score in the SRSF high- or low-score groups of the TCGA cohort. (**D**,**E**) The diversity of immune cell infiltration patterns between patients with various SRSF scores is displayed. (**F**) Correlation analysis between SRSF family genes and infiltrating immune cells. The size of the circle represents the P value, with red representing a positive correlation and green representing a negative correlation. The asterisks represent the statistical *p*-value (* *p* < 0.05, ** *p* < 0.01 and *** *p* < 0.001. ns, no significance).

**Figure 7 cancers-14-04727-f007:**
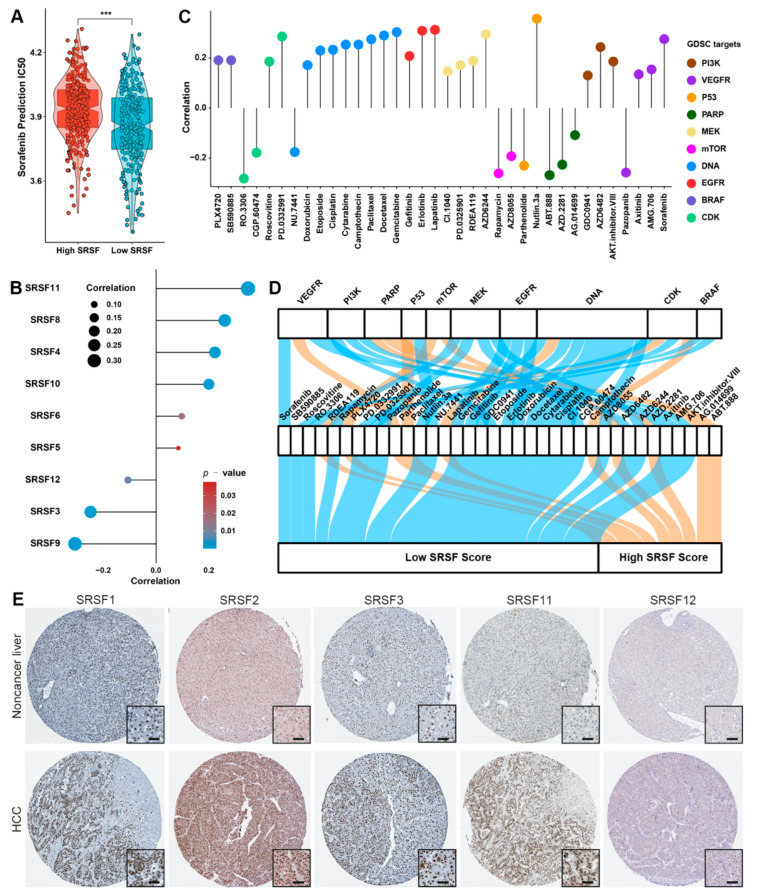
Evaluation of SRSF score for drug sensitivity analysis in HCC patients. (**A**) Predicted IC50 of sorafenib according to the GDSC database. The asterisks represent the statistical *p*-value (*** *p* < 0.001). (**B**) Correlation analysis between SRSF family genes and predicted IC50 of sorafenib from the GDSC database. (**C**) Correlation analysis between SRSF scores and predicted IC50 values for the indicated inhibitors from the GDSC database. (**D**) Alluvial plot showing the changes in drug targets, drug titles and SRSF scores. (**E**) Representative IHC images of SRSF family members in HCC and nontumor liver tissue from the Human Protein Atlas. Scale bar denotes 50 μm.

**Figure 8 cancers-14-04727-f008:**
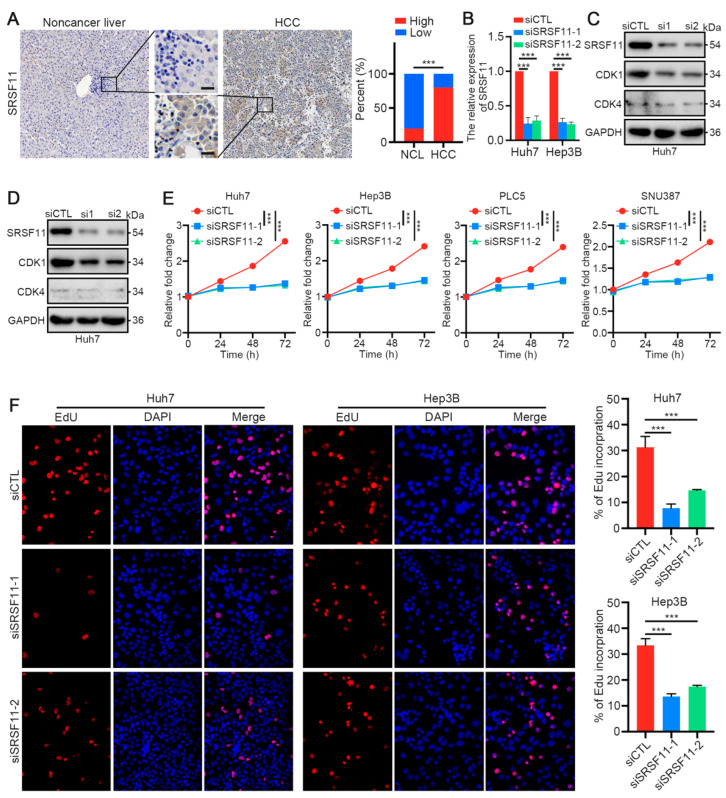
SRSF11 knockdown inhibits the CDK1-dependent cell proliferation in HCC. (**A**) Representative IHC images of SRSF11 in HCC and nontumor liver tissue from our HCC cohort. Scale bar denotes 50 μm. (**B**) qRT-PCR and (**C**,**D**) western blotting for SRSF11, CDK1 and CDK4 in Huh7 or Hep3B cells transfected with SRSF11 siRNAs (siSRSF11) and control siRNA (siCTL), respectively. (**E**) CCK-8 assays for the indicated HCC cells transfected with siSRSF11 or siCTL over a 5-day period. (**F**) EdU assays for the indicated HCC cells transfected with siSRSF11 or siCTL for 48 h. The results are presented as the means ± SDs, and three independent experiments (*N* = 3) were performed in triplicate. Student’s *t*-test was used for statistics. The asterisks represent the statistical *p*-value (*** *p* < 0.001).

**Figure 9 cancers-14-04727-f009:**
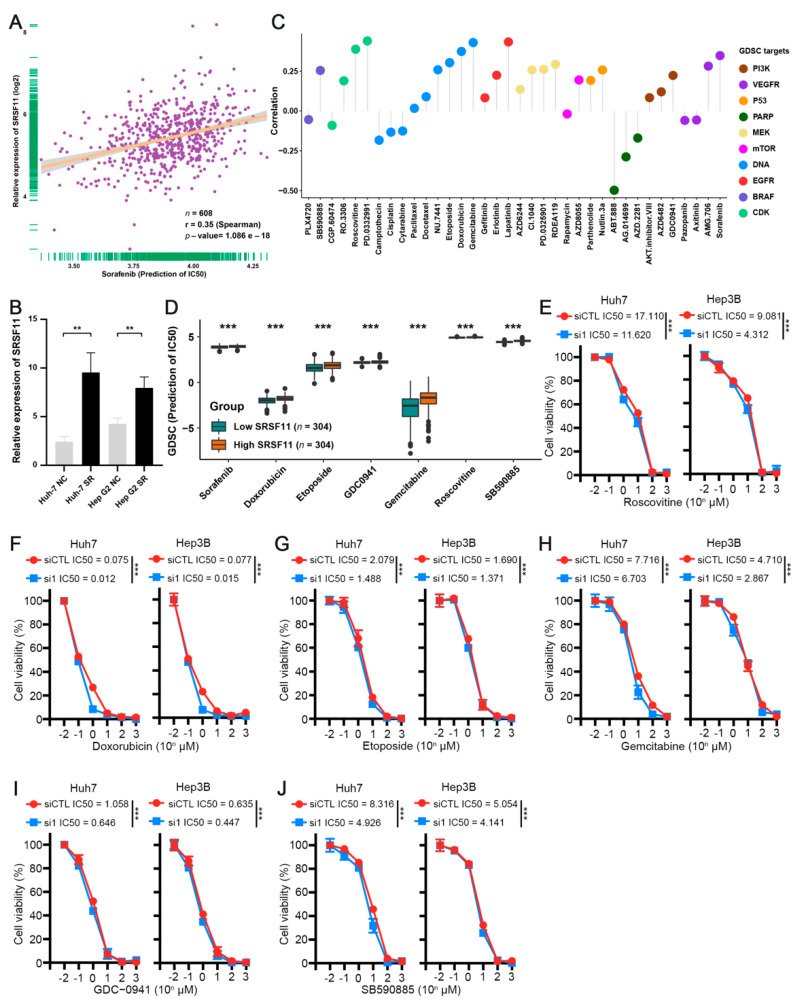
Silencing of SRSF11 improves drug sensitivity to systemic therapy in HCC. (**A**) Correlation analysis between SRSF11 expression and the predicted IC50 of sorafenib. (**B**) The relative expression of SRSF11 mRNA in parental HCC cells and corresponding sorafenib-resistant (SR) HCC cells. (**C**) Correlation analysis between SRSF11 expression and predicted IC50 values for the indicated inhibitors from the GDSC database. (**D**) Predicted IC50 values for the indicated inhibitors are stratified by the SRSF score. (**E**–**J**) CCK-8 assays for Huh7 or Hep3B cells transfected with siSRSF11 and siCTL, respectively, and treated with a range of concentrations of the indicated inhibitors for 3 days. The results are presented as the means ± SDs, and three independent experiments (*N* = 3) were performed in triplicate. Student’s *t*-test was used for statistics. The asterisks represent the statistical *p*-value (** *p* < 0.01 and *** *p* < 0.001).

## Data Availability

Not applicable.

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
