# Peer review of "Comprehensive Molecular Analysis Identified an SRSF Family-Based Score for Prognosis and Therapy Efficiency Prediction in Hepatocellular Carcinoma"

_cancers, 2022, doi:10.3390/cancers14194727_

Round 1

Reviewer 1 Report

Dear Authors, I have read with interest this manuscript that concerns the molecular analysis of SRSF genes in HCC. The paper is overall well written,  the analyses are comprehensive and importantly they are convincing. I would suggest to add some clinico-pathological data of the 30 human cases. While 30 is not a big number, and statistics may be uncertain, I would consider to add some analyses (i.e., association w/ TNM, w/ tumor number, w/ tumor size, w/ vascular invasion, w/ etiology, etc...)

Author Response

Dear Reviewer:

Thank you very much for your letter and comments on our manuscript entitled “Comprehensive molecular analysis identified an SRSF family-based score for prognosis and therapy efficiency prediction in hepatocellular carcinoma”. The comments are very helpful and we have revised our manuscript to incorporate your suggestions as follows:

Point 1: Dear Authors, I have read with interest this manuscript that concerns the molecular analysis of SRSF genes in HCC. The paper is overall well written,  the analyses are comprehensive and importantly they are convincing. I would suggest to add some clinico-pathological data of the 30 human cases. While 30 is not a big number, and statistics may be uncertain, I would consider to add some analyses (i.e., association w/ TNM, w/ tumor number, w/ tumor size, w/ vascular invasion, w/ etiology, etc...)

Response 1: Thanks for your valuable comment. We have added the clinicopathological data of these 30 human cases as Table S1. Additionally, since our sample size is too small, we also added the analysis of SRSF11 expression and HCC clinicopathological parameters of the TCGA-LIHC cohort as Figure S2B.

Reviewer 2 Report

The present study investigated the biomedical significance of serine/arginine-rich splicing factors (SRSF) in hepatocellular carcinoma (HCC). The expression of SRSF family genes showed the significant relationship to the prognosis of HCC patients. The expression of SRSF11 had the biological role in the CDK1-dependent proliferation of HCC cells. The reviewer considers the present study has sufficient number of experiments and the manuscript is well-written. The reviewer would like to ask some queries to the authors as described below.

1.    The authors showed the significant relationship between the expression of SRSF family genes and overall survival of HCC patients. As described in the section of Introduction, the high frequency of metastasis and recurrence is the serious problem in the treatment of HCC. The reviewer would like to know the value of SRSF expression in the prediction of HCC recurrence. Is there any significance between the SRSF expression and the recurrence-free survival of HCC patients?

2.    The reviewer cannot understand the method of data analysis using GTEx, TCGA and ICGC databases. The reviewer considers the authors add some sentences in the section of Methods that describes the method of data analysis and the characteristics of each database (Ex. patient number, etc.).

3.    The reviewer cannot understand why the authors focused on the role of SRSF expression in the drug sensitivity. Is there any evidence that show the biological function of SRSF family genes on drug sensitivity? If so, the reviewer request to add some comments in the section of Introduction or Discussion with some citation.

Author Response

Dear Reviewer:

Thank you very much for your letter and comments on our manuscript entitled “Comprehensive molecular analysis identified an SRSF family-based score for prognosis and therapy efficiency prediction in hepatocellular carcinoma”. The comments are very helpful and we have revised our manuscript to incorporate your suggestions as follows:

The present study investigated the biomedical significance of serine/arginine-rich splicing factors (SRSF) in hepatocellular carcinoma (HCC). The expression of SRSF family genes showed the significant relationship to the prognosis of HCC patients. The expression of SRSF11 had the biological role in the CDK1-dependent proliferation of HCC cells. The reviewer considers the present study has sufficient number of experiments and the manuscript is well-written. The reviewer would like to ask some queries to the authors as described below.

Point 1: The authors showed the significant relationship between the expression of SRSF family genes and overall survival of HCC patients. As described in the section of Introduction, the high frequency of metastasis and recurrence is the serious problem in the treatment of HCC. The reviewer would like to know the value of SRSF expression in the prediction of HCC recurrence. Is there any significance between the SRSF expression and the recurrence-free survival of HCC patients?

Response 1: Thanks for your valuable comment. The HCC data in our study were all obtained from the TCGA and ICGC databases. Unfortunately, the clinical information of these two databases does not contain data on recurrence-free survival, so we were unable to perform this analysis. However, we added the data analysis on progression-free survival (PFS) of HCC patients from the TCGA database, which can also partially demonstrate the relationship between SRSF family genes and HCC recurrence or progression.

Point 2: The reviewer cannot understand the method of data analysis using GTEx, TCGA and ICGC databases. The reviewer considers the authors add some sentences in the section of Methods that describes the method of data analysis and the characteristics of each database (Ex. patient number, etc.).

Response 2: Thanks for your valuable feedback. We have added the methods for three dataset merging in the “Data sources and preprocessing” section of Materials and methods. In addition, we added the characteristics of each database including the patient number.

Point 3: The reviewer cannot understand why the authors focused on the role of SRSF expression in the drug sensitivity. Is there any evidence that show the biological function of SRSF family genes on drug sensitivity? If so, the reviewer request to add some comments in the section of Introduction or Discussion with some citation.

Response 3: We focused on the role of SRSF expression in drug sensitivity for the following reasons: 1) Previous studies have demonstrated that SRSF family genes may be the therapeutic targets for tumors. 2) Our analysis suggested that SRSF11 was significantly related to cell cycle and tumor proliferation. Thus, we would like to further investigate whether SRSF11 expression affects the inhibitory effect of inhibitors of these signaling pathways. In addition, we have added some comments and citations to the fifth paragraph of the Discussion section.
